# Universal Virucidal Activity of Calcium Bicarbonate Mesoscopic Crystals That Provides an Effective and Biosafe Disinfectant

**DOI:** 10.3390/microorganisms10020262

**Published:** 2022-01-24

**Authors:** Rikio Kirisawa, Rika Kato, Koichi Furusaki, Takashi Onodera

**Affiliations:** 1Department of Pathobiology, School of Veterinary Medicine, Rakuno Gakuen University, Ebetsu 069-8501, Japan; s-ak.m-skt.m-g07@docomo.ne.jp; 2Mineral Activation Technical Research Center, Ohmuta 836-0041, Japan; riken@jade.plala.or.jp; 3Research Center for Food Safety, Graduate School of Agricultural and Life Sciences, University of Tokyo, Tokyo 113-8657, Japan; atonode@g.ecc.u-tokyo.ac.jp

**Keywords:** calcium bicarbonate, mesoscopic crystals, animal viruses, disinfectant, foot-and-mouth disease virus, SARS-CoV-2, virucidal activity

## Abstract

We investigated the virucidal effects in solution of a new type of disinfectant, calcium bicarbonate mesoscopic crystals, designated CAC-717, against various types of virus. CAC-717 in solution is alkaline (pH 12.4) and has a self-electromotive force that generates pulsed electrical fields. Upon application to human skin, the pH of the solution becomes 8.4. CAC-717 contains no harmful chemicals and is thus non-irritating and harmless to humans and animals. Its virucidal effects were tested against six types of animal virus: enveloped double-strand (ds)-DNA viruses, non-enveloped ds-DNA viruses, non-enveloped single strand (ss)-DNA viruses, enveloped ss-RNA viruses, non-enveloped ss-RNA viruses, and non-enveloped ds-RNA viruses. The treatment resulted in a reduction in viral titer of at least 3.00 log_10_ to 6.38 log_10_. Fetal bovine serum was added as a representative organic substance. When its concentration was ≥20%, the virucidal effect of CAC-717 was reduced. Real-time PCR revealed that CAC-717 did not reduce the quantity of genomic DNA of most of the DNA viruses, but it greatly reduced that of the genomic RNA of most of the RNA viruses. CAC-717 may therefore be a useful biosafe disinfectant for use against a broad range of viruses.

## 1. Introduction

Control of microorganisms is essential in many fields, such as medicine, veterinary care, dentistry, food processing, and in the environmental protection. Disinfection of pathogens is typically conducted using chemical disinfectants such as aldehydes, iodine, chlorine phenolics, sodium carbonate, citric acid, and acetic acid [1,2,3]. However, there are various problems with the use of such disinfectants. For example, many disinfectants are corrosive or act as irritants, and they may leave a residue [4]. An ideal disinfectant would be broad-spectrum, non-toxic, non-irritating, noncorrosive, and effective for to humans, animals, and the environment. Unfortunately, no such ideal disinfectant has yet been found.

Animal viruses are classified into six groups based on the presence or absence of a viral envelope and the type of viral genome they contain: enveloped double-strand (ds)-DNA, non-enveloped ds-DNA, non-enveloped single-strand (ss)-DNA, enveloped ss-RNA, non-enveloped ss-RNA, and non-enveloped ds-RNA. These groups of viruses vary in their susceptibility to disinfectants. Most viruses exhibit reduced infectivity or are inactivated under highly acidic or alkaline conditions [5,6,7,8,9]. Non-enveloped viruses are more resistant to disinfectants than enveloped viruses are. Small non-enveloped viruses, except those in the genus *Aphthovirus* (in the family *Picornaviridae*), are more resistant to disinfectants than large non-enveloped viruses are [1,10]. Foot-and-mouth disease virus (FMDV), which belongs to the genus *Aphthovirus* and causes one of the most contagious animal diseases (FMD, which is associated with substantial economic losses), is inactivated at pH values <6.0 or >9.0 [11]. The Ministry of Agriculture, Forestry and Fisheries of Japan, therefore, recommends using 2% sodium hydroxide and 4% sodium carbonate solutions together with calcium hydroxide to inactivate the virus. Other small non-enveloped viruses can be inactivated with sodium hypochlorite and glutaraldehyde but not with acids, alcohols, alkalis, biguanides, oxidizing agents, or quaternary ammonium compounds, all of which are effective for inactivating other viruses. Therefore, sodium hypochlorite and glutaraldehyde may be considered universal disinfectants. However, sodium hypochlorite releases chlorine odors and chlorite gas, which corrodes metals and is poisonous for humans and animals [2,3], and glutaraldehyde fumes are irritating and toxic to the skin, mucous membranes, and respiratory tract [3,10]. Alcohol-based disinfectants are used widely but are unsuitable for use in people with skin damage, and long-term use may lead to skin irritation. There is also a risk of fire if the alcohol vaporizes in a confined space.

An entirely new type of disinfectant, a solution of electrically charged calcium bicarbonate mesoscopic crystals (CAC-717), has recently been developed. CAC-717 has been shown to have virucidal effects against influenza A virus and SARS-CoV-2, which are enveloped ss-RNA viruses [12,13], as well as against human norovirus, mouse norovirus [14], and feline calicivirus (FCV) [15], which are non-enveloped ss-RNA viruses. In this study, we expanded our investigation to test the virucidal effects of CAC-717 against a broad range of viruses including all six groups of animal viruses.

## 2. Materials and Methods

### 2.1. Viruses and Cell Cultures

Twenty-two viruses were used in this study (Table 1). We used three serotype FMD viruses—A, O, and Asia1—as well as bovine rhinitis B virus (BRBV) and FCV were used as surrogates for FMDV and human norovirus, respectively [16,17,18]. The experiments using the three FMDV serotypes were conducted in the Department of Biochemistry and Immunology, National Institute of Veterinary Research, Hanoi, Vietnam.

Canine parvovirus 2 (CPV-2), strain 97-008, was provided by the Kyoritsu Seiyaku Corporation, Tsukuba, Japan. Bovine respiratory syncytial virus (BRSV) strain, rs-52, was provided by KYOTOBIKEN, Kyoto, Japan. Equine influenza A virus (EqIV) strain A/Equine/Hayakita/1/2007 is a field strain that was isolated in our laboratory in 2007. SARS-CoV-2 strain JPN/TY/WK-521 was provided by the National Institute of Infectious Diseases, Tokyo, Japan. FCV strain F9 was provided by Dr. T. Tajima, Osaka Prefecture University, Osaka, Japan.

We used Madin–Darby bovine kidney (MDBK) cells for viral propagation and titration of infectious bovine rhinotracheitis virus (IBRV) and bovine parainfluenza virus 3 (BPIV-3). We used ESK cells (a porcine kidney cell line; the National Institute of Animal Health, Japan) to propagate and titrate pseudorabies virus (PrV) and vesicular stomatitis virus (VSV). Madin–Darby canine kidney (MDCK) cells were used to propagate and titrate canine herpesvirus 1 (CHV-1), CPV-2, swine influenza A virus (SwIV), and EqIV. We used primary horse fetal kidney (HFK) cells to propagate and titrate equine herpesvirus 1 (EHV-1), and bovine testis (BT) cells for bovine adenovirus 7 (BAdV-7). For BRSV, we used VeroE6 (African green monkey kidney) cells, and for canine distemper virus (CDV), we used Crandell feline kidney (CRFK)-SLAM cells [19], which express canine signaling lymphocyte activation molecule (SLAM). Quail methylcholanthrene-induced fibrosarcoma (QT-6) (provided by Dr. T. Kobayashi, Osaka University, Osaka, Japan) were used to propagate and titrate bulbul orthoreovirus (BuORV) and Newcastle disease virus (NDV). We used HRT-18G cells (human ileocecal colorectal adenocarcinoma, ATCC CRL-11663; American Type Culture Collection, Manassas, VA, USA) for bovine coronavirus (BCoV). Transmembrane protease, serine 2 (TMPRSS2)-expressing VeroE6 (VeroE6/TMPRSS2) cells [20] (Japanese Collection of Research Bioresources Cell Bank, Ibaraki, Japan), were used to propagate and titrate of SARS-CoV-2, which was handled at a biosafety level 3 facility. We used primary bovine fetal muscle (BFM) cells for bovine viral diarrhea virus (BVDV)-I and BVDV-II, and BHK-21 cells (hamster kidney cell line, Cellosaurus ID: CVCL_1914) for FMDV. Primary bovine fetal kidney (BFK) cells were used to propagate and titrate BRBV, and CRFK cells were used for FCV. We used MA-104 (African green monkey kidney) cells (RIKEN BioResource Center Cell Bank, Tsukuba, Japan) for bovine rotavirus (BRoV).

Each virus suspension was harvested after the appropriate number of days of incubation for that virus and separated from cellular debris by centrifugation at 1500 rpm for 5 min. Small aliquots of the supernatant were then divided into microtubes and stored at −80 °C until used.

**Table 1 microorganisms-10-00262-t001:** Virus strains used in this study.

Envelope	Genome	Family	Genus	Virus	Strain	Reference
+	ds-DNA	*Herpesviridae*	*Varicellovirus*	infectious bovine rhinotracheitis virus (IBRV)	Los Angeles	[21]
				pseudorabies virus (PrV)	MY-1	[22]
				canine herpesvirus 1 (CHV-1)	GCH-1	[23]
				equine herpesvirus 1 (EHV-1)	HH1	[24]
-	ds-DNA	*Adenoviridae*	*Atadenovirus*	bovine adenovirus 7 (BAdV-7)	Fukuroi	[25]
-	ss-DNA	*Parvoviridae*	*Protoparvovirus*	canine parvovirus 2 (CPV-2)	97-008	this study
+	ss-RNA	*Paramyxoviridae*	*Respirovirus*	bovine parainfluenza virus 3 (BPIV-3)	BN-1	[26]
			*Pneumovirus*	bovine respiratory syncytial virus (BRSV)	rs-52	[27]
			*Morbillivirus*	canine distemper virus (CDV)	KDK-1	[28]
			*Avulavirus*	Newcastle disease virus (NDV)	Miyadera	[29]
+	ss-RNA	*Rhabdoviridae*	*Vesiculovirus*	vesicular stomatitis virus (VSV)	New Jersey	[30]
+	ss-RNA	*Coronaviridae*	*Betacoronavirus*	SARS-CoV-2	JPN/TY/WK-521	[31]
				bovine coronavirus (BCoV)	Kakegawa	[32]
+	ss-RNA	*Orthomyxoviridae*	*AlphaInfluenzavirus*	swine influenza A virus (pdm09, H1N1) (SwIV)	A/swine/Ibaraki/46/ 2010	[33]
				equine influenza A virus (H3N8) (EqIV)	A/Equine/Hayakita/ 1/2007	this study
+	ss-RNA	*Flaviviridae*	*Pestivirus*	bovine viral diarrhea virus I (BVDV-I)	Nose	[34]
				bovine viral diarrhea virus II (BVDV-II)	KZ-91CP	[35]
-	ss-RNA	*Picornaviridae*	*Aphthovirus*	foot-and-mouth disease virus (FMDV)	A/Vietnam/HSMD05/2003	this study
					O/Vietnam/HSMD03/2005	this study
					Asia1/Vietnam/HSMD04/2005	this study
				bovine rhinitis B virus (BRBV)	EC11	[36]
-	ss-RNA	*Caliciviridae*	*Vesivirus*	feline calicivirus (FCV)	F9	[37]
-	ds-RNA	*Reoviridae*	*Rotavirus*	bovine rotavirus (BRoV)	strain 22R (G6P [5])	[38]
			*Orthoreovirus*	bulbul orthoreovirus (BuORV)	Pycno-1	[39]

The BHK-21, BT, BFM, HFK, MDBK, Vero, MDCK, CRFK, MA-104, and ESK cells were grown in Eagle’s minimum essential medium (MEM; Life Technologies Japan, Tokyo, Japan), supplemented with 10% inactivated fetal bovine serum (FBS) and antibiotics, at 37 °C under 5% CO_2_. The CRFK-SLAM, QT-6, and HRT-18G cells were grown in Dulbecco’s modified Eagle’s medium (DME; Sigma–Aldrich Japan, Tokyo, Japan), supplemented with 10% FBS, at 37 °C under 5% CO_2_. The VeroE6/TMPRSS2 cells were grown in DME, supplemented with 5% FBS and antibiotics, at 37 °C under 5% CO_2_. A culture medium supplemented with 4% FBS was used as a maintenance medium for all cells except VeroE6/TMPRSS2 cells, for which we used DME supplemented with 2% FBS as a maintenance medium.

### 2.2. Disinfectant

The CAC-717 was provided by Santa Mineral (Tokyo, Japan) for testing as a disinfectant. The CAC-717 solution presents as clear, colorless, odorless water. The preparation procedure is described in detail by Sakudo et al. [40]. Briefly, an electric field is applied to mineral water containing calcium bicarbonate, which is extracted from several plants including family Compositae plants, family Rosaceae plants, *Acer* (leaf part and stem part), *Betura platyphylla var. japonica,* and *Cryptomeria japonica,* to produce the electrically charged material known as CAC-717 [12,40]. A Teflon-coated electrostatic field electrode (N-800N, Mineral Activation Technical Research Center, Ohmuta, Japan; Japan Patent No. 5864010) is used to create the electric field, and a voltage of 2 × 10^4^ V is applied for 48 h. The CAC-717 solution in distilled water (FDA/USA Regulation No. 880.6890, Class 1 disinfectant, Japan Patent No. 5778328) has a pH of about 12.4 and contains calcium hydrogen carbonate particles (1120 mg/L) and carbon complex microparticles (50–500 nm) with a mesoscopic structure [12,40]. For the formation of the mesoscopic structure, the presence of interactions in the binding of calcium hydrogen carbonate to essential elements in the plants such as potassium, iron, manganese, zinc, copper, and chlorine is essential (Furusaki et al., unpublished data). In the process of extraction of calcium bicarbonate from plants, these essential elements were also extracted. However, the concentrations and proportions of the essential elements in the extracts differed depending on the plant. Therefore, plants were chosen and mixed in appropriate proportions to obtain an extraction material that is suitable for making a mesoscopic structure [40]. Since the major component of CAC-717 (calcium bicarbonate) exists only in an aqueous solution and cannot be recovered as a solid substance, the concentration of calcium bicarbonate is estimated based on inductively coupled plasma atomic emission spectroscopy, which provides the calcium concentration, and total organic concentration analyzers, which provide the total carbon concentration, total organic carbon concentration, and total inorganic carbon concentration. We used the CAC-717 in its undiluted form in all experiments.

Rabbit skin toxicity tests found that CAC-717 has no harmful effects. These tests were conducted based on the guidelines adopted by the Ministry of Health, Labour and Welfare of Japan: Biological Evaluation of Medical Devices-Part 10: Tests for Irritation and Skin Sensitization (ISO 10993-10, 2 July 2006) [12]. When CAC-717 is applied to human skin, its pH becomes 8.4 [12]. Furthermore, rabbit eye toxicity tests following the Organisation for Economic Co-operation and Development’s Guidelines for Testing Chemicals No. 405: Acute Eye Irritation/Corrosion under the same animal welfare requirements (ISO 10993-2, 2 July 2006) have indicated that CAC-717 has no harmful effects [12].

### 2.3. Disinfection Procedure

Disinfectant procedures were conducted according to ASTM-E1052.

Nine volumes of CAC-717 were mixed with one volume of virus and incubated for approximately 2 s, 10 s, 30 s, 1 min, 15 min, 30 min, or 60 min at room temperature. CAC-717 was then inactivated by adding one volume of 1.0 M HEPES buffer (pH 7.2). One volume of 9.5% NaCl solution was added to bring the osmotic pressure of the mixture to a physiological condition. As controls, we used tap water and maintenance medium instead of CAC-717. As another control, we used CAC-717 pretreated with one volume of 1.0 M HEPES buffer to confirm its inactivation of CAC-717 activity. In the case of FMDV, the mixture of CAC-717 and virus was incubated for 60 min only. The CAC-717 was removed by gel filtration using a Sephadex LH-20 (GE Healthcare Bio-Science AB, Uppsala, Sweden) equilibrated with PBS [41]. For viral genome extraction, one volume of 1.0 M HEPES buffer was added to the CAC-717-treated virus solution to bring the mixture’s pH to neutral in order to facilitate genome extraction and inactivate the CAC-717 activity. As a control, one volume of HEPES buffer was added to nine volumes of CAC-717 before adding one volume of virus solution. As other controls, distilled water (DW) and maintenance medium were used instead of CAC-717. Viral titers were determined by titration in the appropriate cell cultures, as described below. Each titration was conducted in duplicate. We examined the effects of CAC-717 against viral genetic material by quantifying the number of genome copies using real-time PCR. These experiments were performed twice.

We assumed that a titer reduction of at least four logarithmic steps (4 log_10_) showed that the disinfectant had virus-inactivating properties under the tested conditions, based on to the German Association for the Control of Virus Diseases (DVV) and the Robert Koch Institute (RKI) guidelines [42]. We also assumed, however, that virus inactivation was adequate with a titer reduction of more than three logarithmic steps (3 log_10_) [43].

### 2.4. Viral Infectivity Assay

We used a conventional 50% tissue culture infectious dose (TCID_50_) assay to evaluate the infectivity of all the viruses, using appropriate cell monolayers cultured in a 96-well plate for each virus. Briefly, the cells were inoculated with 25 μL of serially 10-fold-diluted sample. After adsorption for 60 min at 37 °C under 5% CO_2_, 125 μL of maintenance medium was added to each well. The culture medium of cells inoculated with undiluted CAC-717-treated samples was replaced with 150 μL of fresh maintenance medium to ensure the same culture medium conditions in all wells. This procedure was also followed for cells inoculated with undiluted control samples. Next, the cells were incubated at 37 °C under 5% CO_2_, except for cells inoculated with BRBV, BRSV, SwIV, or EqIV, which were incubated at 35 °C. Viral titers were calculated using Behrens–Kärber’s method [44], based on cytopathic effects after culturing for three to seven days, depending on the virus. CPV-2 antigens were detected using an indirect immunofluorescence antibody test with an anti-parvovirus monoclonal antibody (CPV1-2A1, Abcam, Tokyo, Japan) and FITC-conjugated goat anti-mouse IgG (Jackson ImmunoResearch Laboratories, West Grove, PA, USA). Viral titers were determined using Behrens–Kärber’s method, based on the presence or absence of CPV-2 antigen-positive cells under a fluorescence microscope (Olympus, Tokyo, Japan) [45].

### 2.5. Influence of an Organic Substance on the Virucidal Effects of CAC-717

We diluted the CAC-717 two-fold with an equivalent volume of 0, 10, 20, 40, 60, 80, or 100% FBS in DW, and nine volumes of this mixture were added to one volume of IBRV or BAdV-7 and then incubated for 1 min at room temperature. After incubation, we added one volume of 1.0 M HEPES buffer and 9.5% NaCl solution to the virus mixture. As controls, we used maintenance medium and DW instead of the CAC-717 mixture. Viral titers were determined as described above. The titration was conducted in duplicate. The protein concentration was determined using a NanoDrop 2000c spectrophotometer (Thermo Fisher Scientific, Tokyo, Japan) and pH values were determined using a pH meter (D52, Horiba, Kyoto, Japan). All experiments were conducted twice.

### 2.6. DNA and RNA Extraction

We extracted DNA from the DNA viruses using a DNeasy Blood and Tissue kit (Qiagen, Tokyo, Japan) and RNA from the RNA viruses using a Viral RNA Mini kit (Qiagen, Tokyo, Japan). Using random primers, we reverse-transcribed the extracted RNA to cDNA with ReverTra Ace-alpha (Toyobo, Osaka, Japan).

### 2.7. Direct Effect of CAC-717 on Viral Genomes

We mixed nine volumes of CAC-717 with one volume of viral genome extracted from the viral solutions, and incubated this mixture for 60 min at room temperature. We then inactivated the CAC-717 by adding one volume of 1.0 M HEPES buffer (pH 7.2). As a control, we used DW instead of CAC-717, and after incubation of 60 min, added one volume of 1.0 M HEPES buffer. After incubation, we extracted the viral genome from the mixture as described above.

### 2.8. Real-Time PCR

We used a LightCycler 1.5 instrument (Roche Diagnostics, Mannheim, Germany) to amplify the viral DNA or cDNA using SYBR Premix Ex Taq DNA Polymerase (Takara Bio, Kusatsu, Japan) and virus-specific primers (Appendix A). We used DNASIS PRO software (Hitachi Software Engineering Co., Ltd., Tokyo, Japan) to determine the following primer sequences. The sequences BAdV-7-F and BAdV-7-R for BAdV-7 were determined based on its DNA polymerase gene (Genbank U57335.1). The sequences BCoV-N-F and BCoV-N-R for BCoV were determined based on the RNA polymerase gene of the BCoV Kakegawa strain (Genbank AB354579.1). The sequences BRBV-Pol-F and BRBV-Pol-R for BRBV were determined based on the RNA polymerase gene of the BRBV EC11 strain (Genbank EU236594.1). The sequences FCV-RT-F and FCV-RT-R for FCV were determined based on the RNA polymerase gene of the FCV-255 strain (Genbank KM111171.1). The sequences BRoV-G6-VP6-F and BRoV-G6-VP6-R1 for BRoV were selected based on the VP6 gene of the BRoV 22R strain (Genbank AB040055.1). The sequences Pyc-L2-F1 and Pyc-L2-R1 for BuORV were determined based on the L2 segment of the BuORV Pycno-1 strain (Genbank AB914761.1).

Amplification was carried out after denaturation at 95 °C for 1 min followed by 45 cycles of denaturation at 95 °C for 15 s, annealing at the appropriate temperature for the primer pairs (Appendix A) for 30 s, and extension at 72 °C for 12 s. We performed real-time PCR in duplicate for each sample. We conducted the whole experiment twice. We calculated the number of copies of the viral genome using a standard curve plotted with standards containing 10 to 1 × 10^4^ gene copies prepared from the purified PCR product of each virus. The concentration of the purified PCR product was determined using a NanoDrop 2000c spectrophotometer (Thermo Fisher Scientific, Tokyo, Japan).

### 2.9. Statistical Analysis

We calculated the standard error using Microsoft Excel.

## 3. Results

### 3.1. Virucidal Activity of CAC-717

We evaluated the virucidal activity of CAC-717 against 22 viruses (Table 1) after various incubation times ranging from approximately 2 s to 60 min, with the viral solution at room temperature (Table 2 and Table 3). In the case of the FMDV serotypes, we conducted only one experiment, with a 60 min incubation, because we only had access to the laboratory in Vietnam for one week. For the DNA viruses (Table 2), the CAC-717 reduced viral infectivity to below the detection limit in all of the viruses tested after a 2 s incubation period, except for CPV-2, which required a 10 s incubation period to achieve the same result. The virucidal effects are represented at the exponential difference between the viral titers of virus treated with maintenance medium and CAC-717. Treatment with CAC-717 for 10 s resulted in a reduction in viral titer of 3.00 log_10_ to 6.38 log_10_ in enveloped DNA viruses (*Herpesviridae* families) and non-enveloped DVA viruses *(Adenoviridae* and *Parvoviridae* families), depending on the viral titers used in the experiments.

For the RNA (Table 3), the CAC-717 reduced viral infectivity to below the detection limit in all of the viruses tested after a 2 s incubation period, except for NDV, VSV, SwIV, EqIV, BVDV-I, and BVDV-II. The CAC-717 treatment resulted in a reduction of virus titer of 3.00 log_10_ to 6.00 log_10_. For the three FMDV serotypes, the CAC-717 treatment also reduced viral infectivity to below the detection limit, although we only assessed this after a 60 min incubation period. The reduction in viral titer was 3.5 log_10_ to 4.88 log_10_ for these serotypes. In SwIV and EqIV, the CAC-717 treatment reduced their infectivity to below the detection limit after 10 s and 30 s, respectively, and resulted in reductions in their viral titers of at least 4.63 log_10_ and 3.25 log_10_, respectively. In the remaining viruses (NDV, VSV, BVDV-I, and BVDV-II), the CAC-717 treatment reduced their infectivity to below the detection limit after a 30 min incubation period and resulted in reductions in their viral titers of 4.13 log_10_ to 5.50 log_10_. In NDV and VSV, CAC-717 treatment for 15 min resulted in reductions in their viral titers of 3.38 log_10_ and 4.00 log_10_, respectively.

Treatment with tap water for 60 min or CAC-717 that had been pretreated with a 1/9 vol of 1.0 M HEPES buffer (pH 7.2) did not reduce the titer of any of the viruses relative to treatment with maintenance medium.

### 3.2. Influence of an Organic Substance on the Virucidal Effects of CAC-717

We examined the effects of an organic substance (FBS) on the virucidal effects of CAC-717 against IBRV and BAdV-7 (as representative enveloped and non-enveloped viruses, respectively). For the approval of virucidal efficiency based on a suspension test, the DVV and RKI guidelines require a contact time of not less than 30 s. We therefore chose a contact time of 1 min. In the presence of FBS at concentrations of up to 10%, the CAC-717 treatment reduced IBRV infectivity to below the detection limit and resulted in a reduction in the viral titer of at least 6.38 log_10_ relative to treatment with maintenance medium (Table 4). At an FBS concentration of 20%, the CAC-717 treatment resulted in a reduction in the viral titer of 4.75 log_10_. With BAdV-7, at an FBS concentration of 5%, the CAC-717 treatment reduced viral infectivity to below the detection limit and resulted in a decrease in the viral titer of 5.63 log_10_. At an FBS concentration of 10%, the treatment resulted in a reduction in the viral titer of 3.75 log_10_. The pH decreased gradually with increasing FBS concentration, falling below 12 at an FBS concentration of about 20%. Incubation with DW or CAC-717 that had been pretreated with a 1/9 volume of 1.0 M HEPES buffer did not result in a reduction in the titer relative to treatment with maintenance medium.

### 3.3. Effect of CAC-717 on the Viral Genome within the Virion

We assessed the number of copies of viral genomes using real-time PCR with virus-specific primers and compared these between the CAC-717, DW, and maintenance medium treatments (Table 5). We used the same samples (virus-infected culture supernatant) as in the experiments presented in Table 3 and Table 4 to analyze the virucidal activity. The CAC-717 did not significantly reduce the quantity of genomic DNA from the DNA viruses, except for EHV-1, in which it reduced the number of genome copies to <10% of the number of copies extracted from the maintenance medium treatment. In contrast, the number of copies of the RNA virus genomes, except for those of BVDV-I, BVDV-II, and SARS-CoV-2, was drastically reduced by CAC-717 treatment. DW treatment did not destroy the viral genomes.

### 3.4. Direct Effect of CAC-717 on Viral Genomes

To test whether direct application of CAC-717 may affect the viral genomes that were not significantly affected by our standard treatment (Table 5), we applied the CAC-717 directly to the extracted viral genomes of PrV, CPV-2, BVDV-I, and BVDV-II. We compared these against genomes that were treated with direct application of DW. This direct application of CAC-717 effectively destroyed these four viral genomes (Table 6).

### 3.5. Virucidal Effect of CAC-717 after Long-Term Storage

Finally, we assessed the virucidal effect of CAC-717 that had been stored for an extended period at room temperature in the absence of sunlight. We used two lots of CAC-717: lot A (storage period, six years and four months) and lot B (storage period, four years and seven months), and examined their virucidal effects against IBRV, using incubation periods of approximately 2 s to 1 min (Table 7). Treatment with both lots of CAC-717 reduced IBRV infectivity to below the detection limit after incubation for 2 s and resulted in a reduction in the viral titer of at least 6.13 log_10_. The pH values of the solutions had not changed after this prolonged storage, and their virucidal effects against IBRV were almost identical to those in the main experiment (Table 3).

## 4. Discussion

In this study, we used viruses from all six groups of animal viruses. Animal viruses are categorized according to the presence or absence of an envelope, whether the genomic structure is DNA or RNA, and whether the genomic strand is single or double. We showed that the infectivity of all the viruses tested was reduced to below the detection limit by treatment with CAC-717. Based on the requirement in the DVV and RKI guidelines for a titer reduction of at least four logarithmic steps (4 log_10_), CAC-717 exhibits virucidal activity against 17 of the 22 viruses we tested. All five remaining viruses exhibited a titer reduction of at least three logarithmic steps (3 log_10_) when treated with CAC-717. We therefore consider CAC-717 to be adequate for inactivating these five viruses as well.

We confirmed the results of previous studies by showing that CAC-717 inactivates influenza A viruses, SARS-CoV-2, and FCV (a surrogate for human norovirus) [12,13,15]. Nakashima et al. [12] used influenza A viruses A/Aichi/2/68 (H3N2) and A/Swine/Wadayama/5/69 (H3N2), whereas we used A/Swine/Ibaraki/46/2010 (pandemic H1N1) and A/Equine/Hayakita/1/2007 (H3N8). CAC-717 thus inactivates influenza A viruses regardless of their HA and NA subtypes. It also inactivated three serotypes of FMDV, suggesting that it may inactivate all seven known serotypes reported by the World Organisation for Animal Health (OIE). However, we did not check their viral genome counts or viral expression in cells after inoculation with the CAC-717-treated virus. We plan to check these in the near future to investigate whether CAC-717 affects viral attachment to cells, penetration into cells, or both.

CAC-717 contains calcium bicarbonate crystals [12]. These mesoscopic crystals contain self-inducing electrical circuits and release many electrons (e^−^) into the water [12]. Since the mesoscopic crystals are charged with e^−^, they can adsorb H^+^ and convert it into H, and as a result the water contains excess OH^−^. This makes the water alkaline, with a pH of about 12.4. This highly alkaline environment may be one factor that affects the proteins on the viral surface, the envelope, or the capsid in non-enveloped viruses, similarly to other disinfectants that have a highly alkaline pH, although, in their case, that is induced by hazardous chemicals. At a high pH, peptide bonds tend to be hydrolyzed, leading to a conformational change in surface viral proteins. However, Shimakura et al. [14] reported that while CAC-717 inactivated human norovirus, a phosphate buffer with a pH of 12.33 did not. They therefore suggested that, in case of human norovirus, an alkaline pH was not an essential factor in the inactivation process.

The self-inducing electrical circuit in the mesoscopic structure microenvironment in CAC-717 is thought to be generated by a high voltage (>1.20 × 10^6^ V) with a current of <0.06 mA over a distance of 0.5 μm (calculated using the inverse square law and the theory of condensed matter physics; Furusaki et al., unpublished data) [12]. This would emit a terahertz wave, which is a type of electromagnetic wave [12]. This self-inducing electrical circuit is also thought to be a pulsed electric field (PEF). PEFs provide an athermal method for inactivating microorganisms by creating nanometer-sized membrane pores in their membranes [46]. Therefore, direct contact between these mesoscopic structures and viruses in the environment may be another important factor in the inactivation. It is thought that a terahertz wave will not damage DNA, membranes, or a culture of epithelial or human embryonic stem cells [47,48]. The wave is strongly absorbed by intermolecular bonds such as the hydrogen bonds in water and H–N bonds in proteins [49]. As a result, resonances are produced that may loosen the protein binding between viral proteins occur, which also affect the viral surface proteins.

Since we observed virucidal activity of CAC-717 against all the viruses we used in this study, we also examined its effects against their viral genomes within the virions under the same conditions as we used with the crude viruses. Treatment of the DNA viruses with CAC-717 did not destroy the viral DNA within the virions. DNA is stable under alkaline conditions, except when the double strand is being cleaved to form a single strand [50]. Furthermore, it has been reported that PEFs do not destroy DNA inside microorganisms [51]. This explains our findings concerning the effect of CAC-717 on DNA viruses. However, when purified viral DNA was treated directly with CAC-717, it was destroyed. This suggests that direct contact with calcium mesoscopic structures may be essential for destroying DNA. In this study, we used crude viral suspensions. One possibility is that culture media that include FBS, cellular proteins, cellular and viral DNA, and unincorporated viral proteins may affect the function of CAC-717. Another possibility is that direct contact with viral DNA was not possible since the size of the calcium mesoscopic structures (50–500 nm) prohibits their entering viral capsids. We therefore plan to conduct further experiments using purified viruses to clarify the activity of CAC-717 against DNA viruses.

In contrast to the DNA viruses, CAC-717 treatment did destroy the RNA of 12 of the 15 RNA viruses we tested. RNA is unstable in alkaline conditions because bases can easily deprotonate the hydrogen from the hydroxyl group on the 2′-carbon atom, leading to the cleavage of the phosphopentose backbone of RNA [50]. One possible explanation for this RNA destruction is that excess OH^−^ ions entered the virions through pores generated by the CAC-717 treatment and raised the pH of the interior of the virus. Three of the RNA viruses, BVDV-I, BVDV-II, and SARS-CoV-2, were not destroyed by CAC-717 treatment. Direct contact of CAC-717 with purified BVDV-I and BVDV-II RNAs, however, did destroy them. Therefore, the movement of OH^−^ ions into the virions caused by CAC-717 did not sufficiently increase the pH of the interior to destroy the RNA in these viruses. The reason for the difference between these three viruses and the other twelve is not clear. BVDV-I and BVDV-II, which belong to the family *Flaviviridae*, obtain their envelope from the endoplasmic reticulum (ER) membrane and bud into the ER [52]. SARS-CoV-2, which belongs to the family *Coronaviridae*, also obtains its envelope and buds in a similar manner [53]. Since these viruses are associated with serum components and contain cellular debris in their viral envelope [52], these factors may affect CAC-717 activity against the viral surface proteins. We are planning to conduct further experiments using purified RNA viruses to investigate this further, as with DNA viruses.

Addition of the organic substance FBS at a final concentration of ≥30% (protein concentration: ≥10.5 mg/mL) reduced CAC-717 activity against IBRV, and FBS at a final concentration of ≥ 20% (protein concentration: ≥7.0 mg/mL) reduced CAC-717 activity against BAdV-7. However, 10% bovine serum albumin (protein concentration: 10 mg/mL) had no effect on the inactivation of influenza A virus by CAC-717 [12]. FBS contains various proteins other than albumin. Thus, a high concentration of a serum protein may reduce the effect of CAC-717. The effects of serum components on the activity of CAC-717 should therefore be investigated. Furthermore, for real world applicability, further investigation taking into account specific viral shedding rates and real-life situations is required.

Several recent studies have shown that hypochlorous acid solution also has virucidal effects against various microorganisms [54,55,56,57]. However, this solution is less stable under conditions such as exposure to ultraviolet radiation or sunlight, and contact with air or organic materials. It requires storage under cool, dark conditions to maintain its microbicidal activity [54]. Our unpublished data using BRBV and BAdV-7 show that weakly acidic hypochlorous acid solution (pH 5.6) at 50 ppm (the residual concentration of chlorine) exhibited diminished antiviral activity at a final FBS concentration of 0.5%. Hypochlorous acid solution should therefore be used for viral inactivation only in areas free of organic materials. In contrast, CAC-717 retains its viral inactivation capacity under conditions of low concentrations of organic materials, such as ≤10% FBS. Further, our experiments showed that the virucidal activity of CAC-717 did not diminish after storage for at least four years at room temperature in the absence of sunlight. Based on these results, CAC-717 is superior to weakly acidic hypochlorous acidic solution with respect to storage conditions and stability under a range of environmental conditions.

In conclusion, our results suggest that CAC-717 inactivates all types of animal viruses. We therefore consider CAC-717 to be a candidate disinfectant for use in universal viral inactivation without causing irritation or harm to humans, animals, or the environment. Although PEFs and the highly alkaline pH may be the factors that induce its virucidal activity, full elucidation of its virucidal mechanism is yet to be resolved.

## Figures and Tables

**Table 2 microorganisms-10-00262-t002:** Virucidal effects of CAC-717 on various DNA viruses. Viral titer is shown as log_10_ TCID_50_/25 μL (mean ± SE). Values are representative values from two separate experiments.

Viruses	Solution								
	CAC-717						CAC-717 + HEPES ^1^	Tap Water	Maintenance Medium
	2 s	10 s	30 s	1 min	15 min	30 min & 60 min	60 min	60 min	60 min
Enveloped									
*Herpesviridae*									
IBRV	≤0.58 *	≤0.58 *	≤0.58 *	≤0.58 *	≤0.58 *	≤0.58 *	6.71 ± 0.13	6.71 ± 0.13	6.96 ± 0.13
PrV	≤0.58 *	≤0.58 *	≤0.58 *	≤0.58 *	≤0.58 *	≤0.58 *	4.83 ± 0.25	4.83 ± 0.00	4.96 ± 0.13
CHV-1	≤0.58 **	≤0.58 **	≤0.58 **	≤0.58 **	≤0.58 **	≤0.58 **	3.33 ± 0.25	2.96 ± 0.13	3.71 ± 0.13
EHV-1	≤0.58 *	≤0.58 *	≤0.58 *	≤0.58 *	≤0.58 *	≤0.58 *	4.83 ± 0.25	4.96 ± 0.13	4.96 ± 0.13
Non-enveloped									
*Adenoviridae*									
BAdV-7	≤0.58 *	≤0.58 *	≤0.58 *	≤0.58 *	≤0.58 *	≤0.58 *	6.46 ± 0.13	6.33 ± 0.25	6.21 ± 0.13
*Parvoviridae*									
CPV-2	2.96 ± 0.13	≤0.58 **	≤0.58 **	≤0.58 **	≤0.58 **	≤0.58 **	3.71 ± 0.13	3.46 ± 0.13	3.58 ± 0.00

^1^ Nine volumes of CAC-717 were pretreated with one volume of 1.0 M HEPES buffer (pH 7.2). * A reduction in viral titer of ≥4 log_10_ relative to treatment with maintenance medium; ** A reduction in viral titer of ≥3 log_10_ relative to treatment with maintenance medium.

**Table 3 microorganisms-10-00262-t003:** Virucidal effects of CAC-717 on various RNA viruses. Viral titer is shown as log_10_ TCID_50_/25 μL (mean ± SE). Values are representative values from two separate experiments.

Viruses	Solution								
	CAC-717						CAC-717 + HEPES ^1^	Tap Water	Maintenance Medium
	2 s	10 s	30 s	1 min	15 min	30 min and 60 min	60 min	60 min	60 min
Enveloped									
*Paramyxoviridae*									
BPIV-3	≤0.58 *	≤0.58 *	≤0.58 *	≤0.58 *	≤0.58 *	≤0.58 *	6.21 ± 0.38	6.33 ± 0.25	6.58 ± 0.00
BRSV	≤0.58 **	≤0.58 **	≤0.58 **	≤0.58 **	≤0.58 **	≤0.58 **	3.58 ± 0.00	3.58 ± 0.00	3.83 ± 0.00
CDV	≤0.58 **	≤0.58 **	≤0.58 **	≤0.58 **	≤0.58 **	≤0.58 **	3.58 ± 0.25	3.46 ± 0.13	3.58 ± 0.00
NDV	4.21 ± 0.13	4.08 ± 0.00	3.71 ± 0.13	3.46 ± 0.13	2.08 ± 0.00 **	≤0.58 *	5.71 ± 0.13	5.33 ± 0.25	5.46 ± 0.13
*Rhabdoviridae*									
VSV	6.08 ± 0.00	6.25 ± 0.25	6.08 ± 0.00	5.46 ± 0.13	2.08 ± 0.25 *	≤0.58 *	6.33 ± 0.00	5.33 ± 0.00	6.08 ± 0.25
*Coronaviridae*									
SARS-CoV-2	≤0.58 *	≤0.58 *	≤0.58 *	≤0.58 *	≤0.58 *	≤0.58 *	4.46 ± 0.13	4.46 ± 0.13	4.58 ± 0.00
BCoV	≤0.58 *	≤0.5 *8	≤0.58 *	≤0.58 *	≤0.58 *	≤0.58 *	5.08 ± 0.00	5.33 ± 0.25	5.33 ± 0.00
*Orthomyxoviridae*									
SwIV (H1N1)	2.83 ± 0.25	≤0.58 *	≤0.58 *	≤0.58 *	≤0.58 *	≤0.58 *	4.83 ± 0.25	5.08 ± 0.25	5.21 ± 0.13
EqIV (H3N8)	3.08 ± 0.00	1.46 ± 0.13	≤0.58 **	≤0.58 **	≤0.58 **	≤0.58 **	3.33 ± 0.00	3.21 ± 0.13	3.83 ± 0.00
*Flaviviridae*									
BVDV-I	4.58 ± 0.00	4.08 ± 0.00	3.83 ± 0.00	3.83 ± 0.25	1.96 ± 0.13	≤0.58 *	4.21 ± 0.13	4.33 ± 0.00	4.71 ± 0.13
BVDV-II	4.58 ± 0.00	4.33 ± 0.00	4.46 ± 0.13	4.71 ± 0.13	2.46 ± 0.13	≤0.58 *	4.96 ± 0.13	4.71 ± 0.13	4.71 ± 0.13
Non-enveloped									
*Picornaviridae*									
FMDV type A						≤0.5 *^,2^		5.00 ± 0.13	5.38 ± 0.13
type O						≤0.5 **^,2^		4.25± 0.00	4.00 ± 0.00
type Asia 1						≤0.5 *^,2^		4.00± 0.00	4.63 ± 0.13
BRBV	≤0.58 *	≤0.58 *	≤0.58 *	≤0.58 *	≤0.58 *	≤0.58 *	4.21 ± 0.13	4.21 ± 0.13	4.71 ± 0.13
*Caliciviridae*									
FCV	≤0.58 *	≤0.58 *	≤0.58 *	≤0.58 *	≤0.58 *	≤0.58 *	7.08 ± 0.25	6.33 ± 0.00	6.58 ± 0.00
*Reoviridae*									
BRoV	≤0.58 *	≤0.58 *	≤0.58 *	≤0.58 *	≤0.58 *	≤0.58 *	4.83 ± 0.00	4.58 ± 0.00	4.96 ± 0.13
BuORV	≤0.58 **	≤0.58 **	≤0.58 **	≤0.58 **	≤0.58 **	≤0.58 **	3.83 ± 0.00	3.71 ± 0.13	3.96 ± 0.13

^1^ Nine volumes of CAC-717 were pretreated with one volume of 1.0 M HEPES buffer (pH 7.2). ^2^ Viral titers after incubation for 60 min; * A reduction in viral titer of ≥4 log_10_ relative to treatment with maintenance medium; ** A reduction in viral titer of ≥3 log_10_ relative to treatment with maintenance medium.

**Table 4 microorganisms-10-00262-t004:** Effect of FBS on the virucidal effects of CAC-717 on IBRV as a representative enveloped virus and BAdV-7 as a representative non-enveloped virus.

Viruses	FBS Concentration (%) in CAC-717							CAC-717 + HEPES ^1^	DW	Maintenance Medium
	0	5	10	20	30	40	50			
IBRV pH	12.4	12.3	12.2	12.0	11.7	11.3	10.9	7.3	5.9	7.5
viral titer ^2^	≤0.58 *	≤0.58 *	≤0.58 *	2.21 * ± 0.13	6.58 ± 0.25	6.33 ± 0.00	6.83 ± 0.00	7.08 ± 0.25	7.08 ± 0.25	6.96 ± 0.13
BAdV-7 pH	12.3	12.2	12.1	11.8	11.4	11.0	10.5	7.3	5.8	7.5
viral titer	≤0.58 *	≤0.58 *	2.46 ** ± 0.13	5.83 ± 0.25	6.96 ± 0.13	6.33 ± 0.25	6.46 ± 0.13	6.46 ± 0.13	6.46 ± 0.38	6.21 ± 0.13

^1^ Nine volumes of CAC-717 were pretreated with one volume of 1.0 M HEPES buffer (pH 7.2). ^2^ Log_10_ TCID_50_/25 μL (mean ± SE); * A reduction in viral titer of ≥4 log_10_ relative to treatment with maintenance medium; ** A reduction in viral titer of ≥3 log_10_ relative to treatment with maintenance medium.

**Table 5 microorganisms-10-00262-t005:** Genome quantification of DNA and RNA viruses treated with CAC-717 for 60 min.

Genome	Envelope	Family	Viruses	Solution	
				CAC-717	DW
DNA	+	*Herpesviridae*	PrV	68.8 ± 3.5 ^1^	121.4 ± 7.2
			CHV-1	82.4 ± 2.8	96.6 ± 5.0
			IBRV	35.8 ± 4.6	91.4 ± 7.5
			EHV-1	8.7 ± 0.4	106.8 ±0.9
	−	*Parvoviridae*	CPV-2	131.0 ± 12.5	79.6 ± 2.1
		*Adenoviridae*	BAdV-7	70.2 ± 5.6	93.4 ± 2.0
RNA	+	*Paramyxoviridae*	BPIV-3	3.0 ± 0.3	90.6± 2.1
			BRSV	0.0 ± 0.0	114.6 ± 3.4
			CDV	0.0 ± 0.0	102.5 ± 1.2
			NDV	2.9 ± 0.1	119.4 ± 5.4
		*Rhabdoviridae*	VSV	0.0 ± 0.0	105.1 ± 9.9
		*Coronaviridae*	SARS-CoV-2	88.5 ± 0.8	108.7 ± 0.7
			BCoV	2.4 ± 0.3	97.4 ± 4.7
		*Flaviviridae*	BVDV-I	68.4 ± 2.9	109.6 ± 11.8
			BVDV-II	22.0 ± 1.6	100.4 ± 2.5
		*Orthomyxoviridae*	SwIV (H1N1)	6.6 ± 0.1	107.2 ± 0.8
			EqIV (H3N8)	0.3 ± 0.0	40.3 ± 0.4
	−	*Picornaviridae*	BRBV	0.6 ± 0.1	75.0 ± 2.4
		*Caliciviridae*	FCV	0.0 ± 0.0	82.0± 1.3
		*Reoviridae*	BRoV	0.0 ± 0.0	137.1 ± 3.3
			BuORV	0.1 ± 0.0	134.1 ± 3.1

^1.^ Values are the mean ± SE percentage of the copy number of genomes extracted from samples treated with maintenance medium.

**Table 6 microorganisms-10-00262-t006:** Direct application of CAC-717 to genomes of the viruses whose DNA or RNA was not significantly affected by CAC-717 application to virions.

Viral Genomes	CAC-717
PrV DNA	0.07 ± 0.01 ^1^
CPV-2 DNA	0.06 ± 0.01
BVDV-I RNA	0.03 ± 0.00
BVDV-II RNA	0.00 ± 0.00

^1^ Values are the mean ± SE percentage of the copy number of genomes extracted from samples treated with DW.

**Table 7 microorganisms-10-00262-t007:** Virucidal effect of CAC-717 on IBRV after long-term storage at room temperature.

Lot no. (Date of Storage)	Storage Period (Years Months)	Initial pH Value	pH Value after Storage	Log_10_ TCID_50_/25 μL (Mean ± SE)					
				CAC-717				CAC-717 + HEPES ^1^	Maintenance Medium
				2 s	10 s	30 s	1 min	1 min	1 min
A (7 January 2015)	6 y 4 m	12.4	12.4	≤0.58 *	≤0.58 *	≤0.58 *	≤0.58 *	7.21 ± 0.13	7.08 ± 0.25
B (17 October 2016)	4 y 7 m	12.4	12.4	≤0.58 *	≤0.58 *	≤0.58 *	≤0.58 *	6.96 ± 0.13	6.71 ± 0.13

^1^ Nine volumes of CAC-717 were pretreated with one volume of 1.0 M HEPES buffer (pH 7.2). * A reduction in viral titer of ≥4 log_10_ relative to treatment with maintenance medium.

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
