# Peer review of "Universal Virucidal Activity of Calcium Bicarbonate Mesoscopic Crystals That Provides an Effective and Biosafe Disinfectant"

_microorganisms, 2022, doi:10.3390/microorganisms10020262_

Round 1

Reviewer 1 Report

The study: Universal virucidal activity of calcium bicarbonate mesoscopic crystals that provides an effective and biosafe disinfectant, by Kirisawa, Kato, Furusaki and Onodera shows results of experiments made with the aim to verify the virocidal activity of calcium bicarbonate mesoscopic crystals (CAC-717).

In general, the manuscript is well written and the results were clearly presented.

However, the experiments showing reductions of viral titer need to be complemented. In addition there are some interpretations made on the results of CAC-717 activity that need to be better clarified.

Since the TCID50 was determined by plaque assay prior to the experiments used to evaluate the virucidal activity of CAC-717, then it should be useful to describe the viral load of each virus used in the experiments with CAC-717. Likewise, authors should discuss if the viral loads used to verify the virucidal effect of CAC-717 make sense in the real world because in some cases infected individuals discharge very high levels of viral particles in saliva for instance. Besides, even a small number of viral particles that remained after CAC-717 activity can trigger a new infection. It is also important to show if after the virucidal effect of CAC-717 is there any viral nucleic acid in these cell cultures.

Experiments showing the effect of CAC-717 in viral particles are flawed. There is no description on how viral particles were isolated and purified to remove the components used in cell cultivation that eventually could interfere with CAC-717 activity. What is the molecular mechanism that allows CAC-717 to destroy RNA within viral capsid (RNA viruses) and not destroy DNA within the capsid of DNA viruses? The most parsimonious explanation is that something went wrong with these experiments.

Minor concerns:

Some tables should be merged (tables 4 and 5; tables 6 and 7) and put on supplementary material (table 1, table 2).

Lines 19-21: “Fetal bovine serum was added ...” What is the expected effect of FBS to the virucidal activity of CAC-717?

Lines 23-24: CAC-717 has virucidal effect by neutralizing viruses outside cells and not necessarily affecting viral life cycle within cells. Change this sentence to make it clear that CAC-717 is not an antiviral agent.

Line 69: “that are present on Earth”. Exclude this sentence

Line 126: what was the source of CAC-717 ?

Line 167 (2.4. Viral infectivity assay): Not sure if plaque assay was done to determine the TCID50 for each virus and then this TCID50 used in the following experiments to evaluate the activity of CAC-717. Please describe better the infectivity assay and the viral load of each TCID50.

Did you confirm any viral expression after CAC-717 treatment?

Line 275: Why did you use 1min of incubation time?

Line 293 (3.3. Effect of CAC-717 on the viral genome within the virion): Please explain how viral particles were isolated from the cell culture and how organic molecules used in cell cultivation were removed in order to not affect the CAC-717 activity.

Results of treatment of viral particles with CAC-717 are inconsistent.

Lines 376-403: All these mechanisms proposed by the authors lack to explain the CAC-717 activity in nucleic acid within viral capsid between RNA and DNA viruses

Lines 426-430: The conclusion that high alkaline environment and PEFs are the main virucidal mechanisms of CAC-717 is purely speculative, there is no evidence of this in the results presented in this study.

Reviewer 2 Report

The Authors present results of virucidal activity of calcium bicarbonate mesoscopic crystals.  The work is well done. However, I have some important remarks. That the results of these tests can be put into practice the preparation procedure must be precisely described. The information "An electric field was applied to mineral water containing calcium bicarbonate to obtain electrically charged material known as CAC-717" is absolutely inadequate. Temperature, concentrations etc. are missing. The product obtained must be standardized. The user must be given methods, as simple as that,  by which he will be able to determine if it is this product and not another. And whether it has virucidal activity at all times. Especially since the Authors guarantee its extremely long effectivenes. Maintaining the same pH value of an alkaline solution for over 6 years is truly amazing. Moreover,the discussion of the mechanism of virucidal action needs to be in-depth. PEF and high pH do not explain the mechanisms of action.

Round 2

Reviewer 2 Report

The Authors revised the manuscript in line with the Reviewers' comments.

However, one thing shoud be further clarified. To obtain CAC-717 Authors used "mineral water containing calcium bicarbonate, which is extracted from plants". Why is the procedure for obtaining this compound so complicated and so mysterious (ref. 40). What are the reasons for choosing such plants.
